# A Linear-Time Kernel Goodness-of-Fit Test

**Wittawat Jitkrittum**
Gatsby Unit, UCL
wittawatj@gmail.com

**Wenkai Xu**
Gatsby Unit, UCL
wenkaix@gatsby.ucl.ac.uk

**Zoltán Szabó**[*]
CMAP, École Polytechnique
zoltan.szabo@polytechnique.edu

**Kenji Fukumizu**
The Institute of Statistical Mathematics
fukumizu@ism.ac.jp

**Arthur Gretton**[*]
Gatsby Unit, UCL
arthur.gretton@gmail.com

## Abstract

We propose a novel adaptive test of goodness-of-fit, with computational cost linear in the number of samples. We learn the test features that best indicate the differences between observed samples and a reference model, by minimizing the false negative rate. These features are constructed via Stein's method, meaning that it is not necessary to compute the normalising constant of the model. We analyse the asymptotic Bahadur efficiency of the new test, and prove that under a mean-shift alternative, our test always has greater relative efficiency than a previous linear-time kernel test, regardless of the choice of parameters for that test. In experiments, the performance of our method exceeds that of the earlier linear-time test, and matches or exceeds the power of a quadratic-time kernel test. In high dimensions and where model structure may be exploited, our goodness of fit test performs far better than a quadratic-time two-sample test based on the Maximum Mean Discrepancy, with samples drawn from the model.

## 1 Introduction

The goal of goodness of fit testing is to determine how well a model density $p(\mathbf{x})$ fits an observed sample $\mathsf{D} = \{\mathbf{x}_i\}_{i=1}^n \subset \mathcal{X} \subseteq \mathbb{R}^d$ from an unknown distribution $q(\mathbf{x})$. This goal may be achieved via a hypothesis test, where the null hypothesis $H_0\colon p = q$ is tested against $H_1\colon p \neq q$. The problem of testing goodness of fit has a long history in statistics [11], with a number of tests proposed for particular parametric models. Such tests can require space partitioning [18, 3], which works poorly in high dimensions; or closed-form integrals under the model, which may be difficult to obtain, besides in certain special cases [2, 5, 30, 26]. An alternative is to conduct a two-sample test using samples drawn from *both* $p$ and $q$. This approach was taken by [23], using a test based on the (quadratic-time) Maximum Mean Discrepancy [16], however this does not take advantage of the known structure of $p$ (quite apart from the increased computational cost of dealing with samples from $p$).

More recently, measures of discrepancy with respect to a model have been proposed based on Stein's method [21]. A Stein operator for $p$ may be applied to a class of test functions, yielding functions that have zero expectation under $p$. Classes of test functions can include the $W^{2,\infty}$ Sobolev space [14], and reproducing kernel Hilbert spaces (RKHS) [25]. Statistical tests have been proposed by [9, 22] based on classes of Stein transformed RKHS functions, where the test statistic is the norm of the smoothness-constrained function with largest expectation under $q$. We will refer to this statistic as the Kernel Stein Discrepancy (KSD). For consistent tests, it is sufficient to use $C_0$-universal kernels [6, Definition 4.1], as shown by [9, Theorem 2.2], although inverse multiquadric kernels may be preferred if uniform tightness is required [15].[2]

---

[*]Zoltán Szabó's ORCID ID: 0000-0001-6183-7603. Arthur Gretton's ORCID ID: 0000-0003-3169-7624.

[2]Briefly, [15] show that when an exponentiated quadratic kernel is used, a sequence of sets D may be constructed that does not correspond to any $q$, but for which the KSD nonetheless approaches zero. In a statistical testing setting, however, we assume identically distributed samples from $q$, and the issue does not arise.

The minimum variance unbiased estimate of the KSD is a U-statistic, with computational cost quadratic in the number $n$ of samples from $q$. It is desirable to reduce the cost of testing, however, so that larger sample sizes may be addressed. A first approach is to replace the U-statistic with a running average with linear cost, as proposed by [22] for the KSD, but this results in an increase in variance and corresponding decrease in test power. An alternative approach is to construct explicit features of the distributions, whose empirical expectations may be computed in linear time. In the two-sample and independence settings, these features were initially chosen at random by [10, 8, 32]. More recently, features have been constructed explicitly to maximize test power in the two-sample [19] and independence testing [20] settings, resulting in tests that are not only more interpretable, but which can yield performance matching quadratic-time tests.

We propose to construct explicit linear-time features for testing goodness of fit, chosen so as to maximize test power. These features further reveal where the model and data differ, in a readily interpretable way. Our first theoretical contribution is a derivation of the null and alternative distributions for tests based on such features, and a corresponding power optimization criterion. Note that the goodness-of-fit test requires somewhat different strategies to those employed for two-sample and independence testing [19, 20], which become computationally prohibitive in high dimensions for the Stein discrepancy (specifically, the normalization used in prior work to simplify the asymptotics would incur a cost cubic in the dimension $d$ and the number of features in the optimization). Details may be found in Section 3.

Our second theoretical contribution, given in Section 4, is an analysis of the relative Bahadur efficiency of our test vs the linear time test of [22]: this represents the relative rate at which the p-value decreases under $H_1$ as we observe more samples. We prove that our test has greater asymptotic Bahadur efficiency relative to the test of [22], for Gaussian distributions under the mean-shift alternative. This is shown to hold regardless of the bandwidth of the exponentiated quadratic kernel used for the earlier test. The proof techniques developed are of independent interest, and we anticipate that they may provide a foundation for the analysis of relative efficiency of linear-time tests in the two-sample and independence testing domains. In experiments (Section 5), our new linear-time test is able to detect subtle local differences between the density $p(\mathbf{x})$, and the unknown $q(\mathbf{x})$ as observed through samples. We show that our linear-time test constructed based on optimized features has comparable performance to the quadratic-time test of [9, 22], while uniquely providing an explicit visual indication of where the model fails to fit the data.

## 2 Kernel Stein Discrepancy (KSD) Test

We begin by introducing the Kernel Stein Discrepancy (KSD) and associated statistical test, as proposed independently by [9] and [22]. Assume that the data domain is a connected open set $\mathcal{X} \subseteq \mathbb{R}^d$. Consider a Stein operator $T_p$ that takes in a multivariate function $\mathbf{f}(\mathbf{x}) = (f_1(\mathbf{x}), \ldots, f_d(\mathbf{x}))^\top \in \mathbb{R}^d$ and constructs a function $(T_p\mathbf{f})(\mathbf{x}) \colon \mathbb{R}^d \to \mathbb{R}$. The constructed function has the key property that for all $\mathbf{f}$ in an appropriate function class, $\mathbb{E}_{\mathbf{x} \sim q}[(T_p\mathbf{f})(\mathbf{x})] = 0$ if and only if $q = p$. Thus, one can use this expectation as a statistic for testing goodness of fit.

The function class $\mathcal{F}^d$ for the function $\mathbf{f}$ is chosen to be a unit-norm ball in a reproducing kernel Hilbert space (RKHS) in [9, 22]. More precisely, let $\mathcal{F}$ be an RKHS associated with a positive definite kernel $k \colon \mathcal{X} \times \mathcal{X} \to \mathbb{R}$. Let $\phi(\mathbf{x}) = k(\mathbf{x}, \cdot)$ denote a feature map of $k$ so that $k(\mathbf{x}, \mathbf{x}') = \langle \phi(\mathbf{x}), \phi(\mathbf{x}') \rangle_{\mathcal{F}}$. Assume that $f_i \in \mathcal{F}$ for all $i = 1, \ldots, d$ so that $\mathbf{f} \in \mathcal{F} \times \cdots \times \mathcal{F} := \mathcal{F}^d$ where $\mathcal{F}^d$ is equipped with the standard inner product $\langle \mathbf{f}, \mathbf{g} \rangle_{\mathcal{F}^d} := \sum_{i=1}^d \langle f_i, g_i \rangle_{\mathcal{F}}$. The kernelized Stein operator $T_p$ studied in [9] is $(T_p\mathbf{f})(\mathbf{x}) := \sum_{i=1}^d \left( \frac{\partial \log p(\mathbf{x})}{\partial x_i} f_i(\mathbf{x}) + \frac{\partial f_i(\mathbf{x})}{\partial x_i} \right) \overset{(a)}{=} \left\langle \mathbf{f}, \boldsymbol{\xi}_p(\mathbf{x}, \cdot) \right\rangle_{\mathcal{F}^d}$, where at $(a)$ we use the reproducing property of $\mathcal{F}$, i.e., $f_i(\mathbf{x}) = \langle f_i, k(\mathbf{x}, \cdot) \rangle_{\mathcal{F}}$, and that $\frac{\partial k(\mathbf{x}, \cdot)}{\partial x_i} \in \mathcal{F}$ [28, Lemma 4.34], hence $\boldsymbol{\xi}_p(\mathbf{x}, \cdot) := \frac{\partial \log p(\mathbf{x})}{\partial \mathbf{x}} k(\mathbf{x}, \cdot) + \frac{\partial k(\mathbf{x}, \cdot)}{\partial \mathbf{x}}$ is in $\mathcal{F}^d$. We note that the Stein operator presented in [22] is defined such that $(T_p\mathbf{f})(\mathbf{x}) \in \mathbb{R}^d$. This distinction is not crucial and leads to the same goodness-of-fit test. Under appropriate conditions, e.g. that $\lim_{\|\mathbf{x}\| \to \infty} p(\mathbf{x}) f_i(\mathbf{x}) = 0$ for all $i = 1, \ldots, d$, it can be shown using integration by parts that $\mathbb{E}_{\mathbf{x} \sim p}(T_p\mathbf{f})(\mathbf{x}) = 0$ for any $\mathbf{f} \in \mathcal{F}^d$ [9, Lemma 5.1]. Based on the Stein operator, [9, 22] define the kernelized Stein discrepancy as

$$S_p(q) := \sup_{\|\mathbf{f}\|_{\mathcal{F}^d} \le 1} \mathbb{E}_{\mathbf{x} \sim q} \left\langle \mathbf{f}, \boldsymbol{\xi}_p(\mathbf{x}, \cdot) \right\rangle_{\mathcal{F}^d} \overset{(a)}{=} \sup_{\|\mathbf{f}\|_{\mathcal{F}^d} \le 1} \left\langle \mathbf{f}, \mathbb{E}_{\mathbf{x} \sim q} \boldsymbol{\xi}_p(\mathbf{x}, \cdot) \right\rangle_{\mathcal{F}^d} = \|\mathbf{g}(\cdot)\|_{\mathcal{F}^d}, \quad (1)$$

where at $(a)$, $\boldsymbol{\xi}_p(\mathbf{x}, \cdot)$ is Bochner integrable [28, Definition A.5.20] as long as $\mathbb{E}_{\mathbf{x} \sim q} \|\boldsymbol{\xi}_p(\mathbf{x}, \cdot)\|_{\mathcal{F}^d} < \infty$, and $\mathbf{g}(\mathbf{y}) := \mathbb{E}_{\mathbf{x} \sim q} \boldsymbol{\xi}_p(\mathbf{x}, \mathbf{y})$ is what we refer to as the *Stein witness function*. The Stein witness function will play a crucial role in our new test statistic in Section 3. When a $C_0$-universal kernel is used [6, Definition 4.1], and as long as $\mathbb{E}_{\mathbf{x} \sim q} \|\nabla_{\mathbf{x}} \log p(\mathbf{x}) - \nabla_{\mathbf{x}} \log q(\mathbf{x})\|^2 < \infty$, it can be shown that $S_p(q) = 0$ if and only if $p = q$ [9, Theorem 2.2].

The KSD $S_p(q)$ can be written as $S_p^2(q) = \mathbb{E}_{\mathbf{x} \sim q} \mathbb{E}_{\mathbf{x}' \sim q} h_p(\mathbf{x}, \mathbf{x}')$, where $h_p(\mathbf{x}, \mathbf{y}) := \mathbf{s}_p^\top(\mathbf{x}) \mathbf{s}_p(\mathbf{y}) k(\mathbf{x}, \mathbf{y}) + \mathbf{s}_p^\top(\mathbf{y}) \nabla_{\mathbf{x}} k(\mathbf{x}, \mathbf{y}) + \mathbf{s}_p^\top(\mathbf{x}) \nabla_{\mathbf{y}} k(\mathbf{x}, \mathbf{y}) + \sum_{i=1}^d \frac{\partial^2 k(\mathbf{x}, \mathbf{y})}{\partial x_i \partial y_i}$, and $\mathbf{s}_p(\mathbf{x}) := \nabla_{\mathbf{x}} \log p(\mathbf{x})$ is a column vector. An unbiased empirical estimator of $S_p^2(q)$, denoted by $\widehat{S^2} = \frac{2}{n(n-1)} \sum_{i<j} h_p(\mathbf{x}_i, \mathbf{x}_j)$ [22, Eq. 14], is a degenerate U-statistic under $H_0$. For the goodness-of-fit test, the rejection threshold can be computed by a bootstrap procedure. All these properties make $\widehat{S^2}$ a very flexible criterion to detect the discrepancy of $p$ and $q$: in particular, it can be computed even if $p$ is known only up to a normalization constant. Further studies on nonparametric Stein operators can be found in [25, 14].

**Linear-Time Kernel Stein (LKS) Test** Computation of $\widehat{S^2}$ costs $\mathcal{O}(n^2)$. To reduce this cost, a linear-time (i.e., $\mathcal{O}(n)$) estimator based on an incomplete U-statistic is proposed in [22, Eq. 17], given by $\widehat{S_l^2} := \frac{2}{n} \sum_{i=1}^{n/2} h_p(\mathbf{x}_{2i-1}, \mathbf{x}_{2i})$, where we assume $n$ is even for simplicity. Empirically [22] observed that the linear-time estimator performs much worse (in terms of test power) than the quadratic-time U-statistic estimator, agreeing with our findings presented in Section 5.

## 3 New Statistic: The Finite Set Stein Discrepancy (FSSD)

Although shown to be powerful, the main drawback of the KSD test is its high computational cost of $\mathcal{O}(n^2)$. The LKS test is one order of magnitude faster. Unfortunately, the decrease in the test power outweighs the computational gain [22]. We therefore seek a variant of the KSD statistic that can be computed in linear time, and whose test power is comparable to the KSD test.

**Key Idea** The fact that $S_p(q) = 0$ if and only if $p = q$ implies that $\mathbf{g}(\mathbf{v}) = \mathbf{0}$ for all $\mathbf{v} \in \mathcal{X}$ if and only if $p = q$, where $\mathbf{g}$ is the Stein witness function in (1). One can see $\mathbf{g}$ as a function witnessing the differences of $p, q$, in such a way that $|g_i(\mathbf{v})|$ is large when there is a discrepancy in the region around $\mathbf{v}$, as indicated by the $i^{th}$ output of $\mathbf{g}$. The test statistic of [22, 9] is essentially given by the degree of "flatness" of $\mathbf{g}$ as measured by the RKHS norm $\|\cdot\|_{\mathcal{F}^d}$. The core of our proposal is to use a different measure of flatness of $\mathbf{g}$ which can be computed in linear time.

The idea is to use a real analytic kernel $k$ which makes $g_1, \ldots, g_d$ real analytic. If $g_i \neq 0$ is an analytic function, then the Lebesgue measure of the set of roots $\{\mathbf{x} \mid g_i(\mathbf{x}) = 0\}$ is zero [24]. This property suggests that one can evaluate $g_i$ at a finite set of locations $V = \{\mathbf{v}_1, \ldots, \mathbf{v}_J\}$, drawn from a distribution with a density (w.r.t. the Lebesgue measure). If $g_i \neq 0$, then almost surely $g_i(\mathbf{v}_1), \ldots, g_i(\mathbf{v}_J)$ will not be zero. This idea was successfully exploited in recently proposed linear-time tests of [8] and [19, 20]. Our new test statistic based on this idea is called the Finite Set Stein Discrepancy (FSSD) and is given in Theorem 1. All proofs are given in the appendix.

**Theorem 1** (The Finite Set Stein Discrepancy (FSSD)). *Let $V = \{\mathbf{v}_1, \ldots, \mathbf{v}_J\} \subset \mathbb{R}^d$ be random vectors drawn i.i.d. from a distribution $\eta$ which has a density. Let $\mathcal{X}$ be a connected open set in $\mathbb{R}^d$. Define $\mathrm{FSSD}_p^2(q) := \frac{1}{dJ} \sum_{i=1}^d \sum_{j=1}^J g_i^2(\mathbf{v}_j)$. Assume that 1) $k \colon \mathcal{X} \times \mathcal{X} \to \mathbb{R}$ is $C_0$-universal [6, Definition 4.1] and real analytic i.e., for all $\mathbf{v} \in \mathcal{X}$, $f(\mathbf{x}) := k(\mathbf{x}, \mathbf{v})$ is a real analytic function on $\mathcal{X}$. 2) $\mathbb{E}_{\mathbf{x} \sim q} \mathbb{E}_{\mathbf{x}' \sim q} h_p(\mathbf{x}, \mathbf{x}') < \infty$. 3) $\mathbb{E}_{\mathbf{x} \sim q} \|\nabla_{\mathbf{x}} \log p(\mathbf{x}) - \nabla_{\mathbf{x}} \log q(\mathbf{x})\|^2 < \infty$. 4) $\lim_{\|\mathbf{x}\| \to \infty} p(\mathbf{x}) \mathbf{g}(\mathbf{x}) = 0$.*

*Then, for any $J \geq 1$, $\eta$-almost surely $\mathrm{FSSD}_p^2(q) = 0$ if and only if $p = q$.*

This measure depends on a set of $J$ test locations (or features) $\{\mathbf{v}_i\}_{i=1}^J$ used to evaluate the Stein witness function, where $J$ is fixed and is typically small. A kernel which is $C_0$-universal and real analytic is the Gaussian kernel $k(\mathbf{x}, \mathbf{y}) = \exp\left(-\frac{\|\mathbf{x}-\mathbf{y}\|_2^2}{2\sigma_k^2}\right)$ (see [20, Proposition 3] for the result on analyticity). Throughout this work, we will assume all the conditions stated in Theorem 1, and consider only the Gaussian kernel. Besides the requirement that the kernel be real and analytic, the remaining conditions in Theorem 1 are the same as given in [9, Theorem 2.2]. Note that if the

FSSD is to be employed in a setting otherwise than testing, for instance to obtain pseudo-samples converging to $p$, then stronger conditions may be needed [15].

## 3.1 Goodness-of-Fit Test with the FSSD Statistic

Given a significance level $\alpha$ for the goodness-of-fit test, the test can be constructed so that $H_0$ is rejected when $n\widehat{\text{FSSD}^2} > T_\alpha$, where $T_\alpha$ is the rejection threshold (critical value), and $\widehat{\text{FSSD}^2}$ is an empirical estimate of $\text{FSSD}_p^2(q)$. The threshold which guarantees that the type-I error (i.e., the probability of rejecting $H_0$ when it is true) is bounded above by $\alpha$ is given by the $(1-\alpha)$-quantile of the null distribution i.e., the distribution of $n\widehat{\text{FSSD}^2}$ under $H_0$. In the following, we start by giving the expression for $\widehat{\text{FSSD}^2}$, and summarize its asymptotic distributions in Proposition 2.

Let $\mathbf{\Xi}(\mathbf{x}) \in \mathbb{R}^{d \times J}$ such that $[\mathbf{\Xi}(\mathbf{x})]_{i,j} = \xi_{p,i}(\mathbf{x}, \mathbf{v}_j)/\sqrt{dJ}$. Define $\boldsymbol{\tau}(\mathbf{x}) := \text{vec}(\mathbf{\Xi}(\mathbf{x})) \in \mathbb{R}^{dJ}$ where $\text{vec}(\mathbf{M})$ concatenates columns of the matrix $\mathbf{M}$ into a column vector. We note that $\boldsymbol{\tau}(\mathbf{x})$ depends on the test locations $V = \{\mathbf{v}_j\}_{j=1}^J$. Let $\Delta(\mathbf{x}, \mathbf{y}) := \boldsymbol{\tau}(\mathbf{x})^\top \boldsymbol{\tau}(\mathbf{y}) = \text{tr}(\mathbf{\Xi}(\mathbf{x})^\top \mathbf{\Xi}(\mathbf{y}))$. Given an i.i.d. sample $\{\mathbf{x}_i\}_{i=1}^n \sim q$, a consistent, unbiased estimator of $\text{FSSD}_p^2(q)$ is

$$\widehat{\text{FSSD}^2} = \frac{1}{dJ} \sum_{l=1}^d \sum_{m=1}^J \frac{1}{n(n-1)} \sum_{i=1}^n \sum_{j \neq i} \xi_{p,l}(\mathbf{x}_i, \mathbf{v}_m) \xi_{p,l}(\mathbf{x}_j, \mathbf{v}_m) = \frac{2}{n(n-1)} \sum_{i<j} \Delta(\mathbf{x}_i, \mathbf{x}_j), \quad (2)$$

which is a one-sample second-order U-statistic with $\Delta$ as its U-statistic kernel [27, Section 5.1.1]. Being a U-statistic, its asymptotic distribution can easily be derived. We use $\xrightarrow{d}$ to denote convergence in distribution.

**Proposition 2** (Asymptotic distributions of $\widehat{\text{FSSD}^2}$). *Let* $Z_1, \ldots, Z_{dJ} \overset{i.i.d.}{\sim} \mathcal{N}(0,1)$. *Let* $\boldsymbol{\mu} := \mathbb{E}_{\mathbf{x} \sim q}[\boldsymbol{\tau}(\mathbf{x})]$, $\boldsymbol{\Sigma}_r := \text{cov}_{\mathbf{x} \sim r}[\boldsymbol{\tau}(\mathbf{x})] \in \mathbb{R}^{dJ \times dJ}$ *for* $r \in \{p, q\}$, *and* $\{\omega_i\}_{i=1}^{dJ}$ *be the eigenvalues of* $\boldsymbol{\Sigma}_p = \mathbb{E}_{\mathbf{x} \sim p}[\boldsymbol{\tau}(\mathbf{x})\boldsymbol{\tau}^\top(\mathbf{x})]$. *Assume that* $\mathbb{E}_{\mathbf{x} \sim q}\mathbb{E}_{\mathbf{y} \sim q}\Delta^2(\mathbf{x}, \mathbf{y}) < \infty$. *Then, for any realization of* $V = \{\mathbf{v}_j\}_{j=1}^J$, *the following statements hold.*

1. *Under* $H_0 : p = q$, $n\widehat{\text{FSSD}^2} \xrightarrow{d} \sum_{i=1}^{dJ}(Z_i^2 - 1)\omega_i$.

2. *Under* $H_1 : p \neq q$, *if* $\sigma_{H_1}^2 := 4\boldsymbol{\mu}^\top \boldsymbol{\Sigma}_q \boldsymbol{\mu} > 0$, *then* $\sqrt{n}(\widehat{\text{FSSD}^2} - \text{FSSD}^2) \xrightarrow{d} \mathcal{N}(0, \sigma_{H_1}^2)$.

*Proof.* Recognizing that (2) is a degenerate U-statistic, the results follow directly from [27, Section 5.5.1, 5.5.2]. $\qquad\square$

Claims 1 and 2 of Proposition 2 imply that under $H_1$, the test power (i.e., the probability of correctly rejecting $H_1$) goes to 1 asymptotically, if the threshold $T_\alpha$ is defined as above. In practice, simulating from the asymptotic null distribution in Claim 1 can be challenging, since the plug-in estimator of $\boldsymbol{\Sigma}_p$ requires a sample from $p$, which is not available. A straightforward solution is to draw sample from $p$, either by assuming that $p$ can be sampled easily or by using a Markov chain Monte Carlo (MCMC) method, although this adds an additional computational burden to the test procedure. A more subtle issue is that when dependent samples from $p$ are used in obtaining the test threshold, the test may become more conservative than required for i.i.d. data [7]. An alternative approach is to use the plug-in estimate $\hat{\boldsymbol{\Sigma}}_q$ instead of $\boldsymbol{\Sigma}_p$. The covariance matrix $\hat{\boldsymbol{\Sigma}}_q$ can be directly computed from the data. This is the approach we take. Theorem 3 guarantees that the replacement of the covariance in the computation of the asymptotic null distribution still yields a consistent test. We write $\mathbb{P}_{H_1}$ for the distribution of $n\widehat{\text{FSSD}^2}$ under $H_1$.

**Theorem 3.** *Let* $\hat{\boldsymbol{\Sigma}}_q := \frac{1}{n} \sum_{i=1}^n \boldsymbol{\tau}(\mathbf{x}_i)\boldsymbol{\tau}^\top(\mathbf{x}_i) - [\frac{1}{n}\sum_{i=1}^n \boldsymbol{\tau}(\mathbf{x}_i)][\frac{1}{n}\sum_{j=1}^n \boldsymbol{\tau}(\mathbf{x}_j)]^\top$ *with* $\{\mathbf{x}_i\}_{i=1}^n \sim q$. *Suppose that the test threshold* $T_\alpha$ *is set to the* $(1-\alpha)$-*quantile of the distribution of* $\sum_{i=1}^{dJ}(Z_i^2-1)\hat{\nu}_i$ *where* $\{Z_i\}_{i=1}^{dJ} \overset{i.i.d.}{\sim} \mathcal{N}(0,1)$, *and* $\hat{\nu}_1, \ldots, \hat{\nu}_{dJ}$ *are eigenvalues of* $\hat{\boldsymbol{\Sigma}}_q$. *Then, under* $H_0$, *asymptotically the false positive rate is* $\alpha$. *Under* $H_1$, *for* $\{\mathbf{v}_j\}_{j=1}^J$ *drawn from a distribution with a density, the test power* $\mathbb{P}_{H_1}(n\widehat{\text{FSSD}^2} > T_\alpha) \to 1$ *as* $n \to \infty$.

*Remark* 1. The proof of Theorem 3 relies on two facts. First, under $H_0$, $\hat{\boldsymbol{\Sigma}}_q = \hat{\boldsymbol{\Sigma}}_p$ i.e., the plug-in estimate of $\boldsymbol{\Sigma}_p$. Thus, under $H_0$, the null distribution approximated with $\hat{\boldsymbol{\Sigma}}_q$ is asymptotically

correct, following the convergence of $\hat{\Sigma}_p$ to $\Sigma_p$. Second, the rejection threshold obtained from the approximated null distribution is asymptotically constant. Hence, under $H_1$, claim 2 of Proposition 2 implies that $n\widehat{\mathrm{FSSD}^2} \xrightarrow{d} \infty$ as $n \to \infty$, and consequently $\mathbb{P}_{H_1}(n\widehat{\mathrm{FSSD}^2} > T_\alpha) \to 1$.

## 3.2 Optimizing the Test Parameters

Theorem 1 guarantees that the population quantity $\mathrm{FSSD}^2 = 0$ if and only if $p = q$ for any choice of $\{\mathbf{v}_i\}_{i=1}^J$ drawn from a distribution with a density. In practice, we are forced to rely on the empirical $\widehat{\mathrm{FSSD}^2}$, and some test locations will give a higher detection rate (i.e., test power) than others for finite $n$. Following the approaches of [17, 20, 19, 29], we choose the test locations $V = \{\mathbf{v}_j\}_{j=1}^J$ and kernel bandwidth $\sigma_k^2$ so as to maximize the test power i.e., the probability of rejecting $H_0$ when it is false. We first give an approximate expression for the test power when $n$ is large.

**Proposition 4** (Approximate test power of $n\widehat{\mathrm{FSSD}^2}$). *Under $H_1$, for large $n$ and fixed $r$, the test power $\mathbb{P}_{H_1}(n\widehat{\mathrm{FSSD}^2} > r) \approx 1 - \Phi\left(\frac{r}{\sqrt{n}\sigma_{H_1}} - \sqrt{n}\frac{\mathrm{FSSD}^2}{\sigma_{H_1}}\right)$, where $\Phi$ denotes the cumulative distribution function of the standard normal distribution, and $\sigma_{H_1}$ is defined in Proposition 2.*

*Proof.* $\mathbb{P}_{H_1}(n\widehat{\mathrm{FSSD}^2} > r) = \mathbb{P}_{H_1}(\widehat{\mathrm{FSSD}^2} > r/n) = \mathbb{P}_{H_1}\left(\sqrt{n}\frac{\widehat{\mathrm{FSSD}^2} - \mathrm{FSSD}^2}{\sigma_{H_1}} > \sqrt{n}\frac{r/n - \mathrm{FSSD}^2}{\sigma_{H_1}}\right)$. For sufficiently large $n$, the alternative distribution is approximately normal as given in Proposition 2. It follows that $\mathbb{P}_{H_1}(n\widehat{\mathrm{FSSD}^2} > r) \approx 1 - \Phi\left(\frac{r}{\sqrt{n}\sigma_{H_1}} - \sqrt{n}\frac{\mathrm{FSSD}^2}{\sigma_{H_1}}\right)$. $\square$

Let $\boldsymbol{\zeta} := \{V, \sigma_k^2\}$ be the collection of all tuning parameters. Assume that $n$ is sufficiently large. Following the same argument as in [29], in $\frac{r}{\sqrt{n}\sigma_{H_1}} - \sqrt{n}\frac{\mathrm{FSSD}^2}{\sigma_{H_1}}$, we observe that the first term $\frac{r}{\sqrt{n}\sigma_{H_1}} = \mathcal{O}(n^{-1/2})$ going to 0 as $n \to \infty$, while the second term $\sqrt{n}\frac{\mathrm{FSSD}^2}{\sigma_{H_1}} = \mathcal{O}(n^{1/2})$, dominating the first for large $n$. Thus, the best parameters that maximize the test power are given by $\boldsymbol{\zeta}^* = \arg\max_{\boldsymbol{\zeta}} \mathbb{P}_{H_1}(n\widehat{\mathrm{FSSD}^2} > T_\alpha) \approx \arg\max_{\boldsymbol{\zeta}} \frac{\mathrm{FSSD}^2}{\sigma_{H_1}}$. Since $\mathrm{FSSD}^2$ and $\sigma_{H_1}$ are unknown, we divide the sample $\{\mathbf{x}_i\}_{i=1}^n$ into two disjoint training and test sets, and use the training set to compute $\frac{\widehat{\mathrm{FSSD}^2}}{\hat{\sigma}_{H_1} + \gamma}$, where a small regularization parameter $\gamma > 0$ is added for numerical stability. The goodness-of-fit test is performed on the test set to avoid overfitting. The idea of splitting the data into training and test sets to learn good features for hypothesis testing was successfully used in [29, 20, 19, 17].

To find a local maximum of $\frac{\widehat{\mathrm{FSSD}^2}}{\hat{\sigma}_{H_1} + \gamma}$, we use gradient ascent for its simplicity. The initial points of $\{\mathbf{v}_i\}_{i=1}^J$ are set to random draws from a normal distribution fitted to the training data, a heuristic we found to perform well in practice. The objective is non-convex in general, reflecting many possible ways to capture the differences of $p$ and $q$. The regularization parameter $\gamma$ is not tuned, and is fixed to a small constant. Assume that $\nabla_{\mathbf{x}} \log p(\mathbf{x})$ costs $\mathcal{O}(d^2)$ to evaluate. Computing $\nabla_{\boldsymbol{\zeta}} \frac{\widehat{\mathrm{FSSD}^2}}{\hat{\sigma}_{H_1} + \gamma}$ costs $\mathcal{O}(d^2 J^2 n)$. The computational complexity of $n\widehat{\mathrm{FSSD}^2}$ and $\hat{\sigma}_{H_1}^2$ is $\mathcal{O}(d^2 Jn)$. Thus, finding a local optimum via gradient ascent is still linear-time, for a fixed maximum number of iterations. Computing $\hat{\Sigma}_q$ costs $\mathcal{O}(d^2 J^2 n)$, and obtaining all the eigenvalues of $\hat{\Sigma}_q$ costs $\mathcal{O}(d^3 J^3)$ (required only once). If the eigenvalues decay to zero sufficiently rapidly, one can approximate the asymptotic null distribution with only a few eigenvalues. The cost to obtain the largest few eigenvalues alone can be much smaller.

*Remark 2.* Let $\hat{\boldsymbol{\mu}} := \frac{1}{n}\sum_{i=1}^n \boldsymbol{\tau}(\mathbf{x}_i)$. It is possible to normalize the FSSD statistic to get a new statistic $\hat{\lambda}_n := n\hat{\boldsymbol{\mu}}^\top (\hat{\Sigma}_q + \gamma\mathbf{I})^{-1}\hat{\boldsymbol{\mu}}$ where $\gamma \geq 0$ is a regularization parameter that goes to 0 as $n \to \infty$. This was done in the case of the ME (mean embeddings) statistic of [8, 19]. The asymptotic null distribution of this statistic takes the convenient form of $\chi^2(dJ)$ (independent of $p$ and $q$), eliminating the need to obtain the eigenvalues of $\hat{\Sigma}_q$. It turns out that the test power criterion for tuning the parameters in this case is the statistic $\hat{\lambda}_n$ itself. However, the optimization is computationally expensive as $(\hat{\Sigma}_q + \gamma\mathbf{I})^{-1}$ (costing $\mathcal{O}(d^3 J^3)$) needs to be reevaluated in each gradient ascent iteration. This is not needed in our proposed FSSD statistic.

# 4 Relative Efficiency and Bahadur Slope

Both the linear-time kernel Stein (LKS) and FSSD tests have the same computational cost of $\mathcal{O}(d^2 n)$, and are consistent, achieving maximum power of 1 as $n \to \infty$ under $H_1$. It is thus of theoretical interest to understand which test is more sensitive in detecting the differences of $p$ and $q$. This can be quantified by the *Bahadur slope* of the test [1]. Two given tests can then be compared by computing the *Bahadur efficiency* (Theorem 7) which is given by the ratio of the slopes of the two tests. We note that the constructions and techniques in this section may be of independent interest, and can be generalised to other statistical testing settings.

We start by introducing the concept of Bahadur slope for a general test, following the presentation of [12, 13]. Consider a hypothesis testing problem on a parameter $\theta$. The test proposes a null hypothesis $H_0 : \theta \in \Theta_0$ against the alternative hypothesis $H_1 : \theta \in \Theta \backslash \Theta_0$, where $\Theta, \Theta_0$ are arbitrary sets. Let $T_n$ be a test statistic computed from a sample of size $n$, such that large values of $T_n$ provide an evidence to reject $H_0$. We use plim to denote convergence in probability, and write $\mathbb{E}_r$ for $\mathbb{E}_{\mathbf{x} \sim r} \mathbb{E}_{\mathbf{x}' \sim r}$.

**Approximate Bahadur Slope (ABS)** For $\theta_0 \in \Theta_0$, let the asymptotic null distribution of $T_n$ be $F(t) = \lim_{n \to \infty} P_{\theta_0}(T_n < t)$, where we assume that the CDF ($F$) is continuous and common to all $\theta_0 \in \Theta_0$. The continuity of $F$ will be important later when Theorem 9 and 10 are used to compute the slopes of LKS and FSSD tests. Assume that there exists a continuous strictly increasing function $\rho : (0, \infty) \to (0, \infty)$ such that $\lim_{n \to \infty} \rho(n) = \infty$, and that $-2 \operatorname{plim}_{n \to \infty} \frac{\log(1 - F(T_n))}{\rho(n)} = c(\theta)$ where $T_n \sim P_\theta$, for some function $c$ such that $0 < c(\theta_A) < \infty$ for $\theta_A \in \Theta \backslash \Theta_0$, and $c(\theta_0) = 0$ when $\theta_0 \in \Theta_0$. The function $c(\theta)$ is known as the *approximate Bahadur slope* (ABS) of the sequence $T_n$. The quantifier "approximate" comes from the use of the asymptotic null distribution instead of the exact one [1]. Intuitively the slope $c(\theta_A)$, for $\theta_A \in \Theta \backslash \Theta_0$, is the rate of convergence of p-values (i.e., $1 - F(T_n)$) to 0, as $n$ increases. The higher the slope, the faster the p-value vanishes, and thus the lower the sample size required to reject $H_0$ under $\theta_A$.

**Approximate Bahadur Efficiency** Given two sequences of test statistics, $T_n^{(1)}$ and $T_n^{(2)}$ having the same $\rho(n)$ (see Theorem 10), the approximate Bahadur efficiency of $T_n^{(1)}$ relative to $T_n^{(2)}$ is defined as $E(\theta_A) := c^{(1)}(\theta_A)/c^{(2)}(\theta_A)$ for $\theta_A \in \Theta \backslash \Theta_0$. If $E(\theta_A) > 1$, then $T_n^{(1)}$ is asymptotically more efficient than $T_n^{(2)}$ in the sense of Bahadur, for the particular problem specified by $\theta_A \in \Theta \backslash \Theta_0$. We now give approximate Bahadur slopes for two sequences of linear time test statistics: the proposed $n\widehat{\text{FSSD}^2}$, and the LKS test statistic $\sqrt{n}\widehat{S_l^2}$ discussed in Section 2.

**Theorem 5.** *The approximate Bahadur slope of $n\widehat{\text{FSSD}^2}$ is $c^{(\text{FSSD})} := \text{FSSD}^2/\omega_1$, where $\omega_1$ is the maximum eigenvalue of $\Sigma_p := \mathbb{E}_{\mathbf{x} \sim p}[\boldsymbol{\tau}(\mathbf{x})\boldsymbol{\tau}^\top(\mathbf{x})]$ and $\rho(n) = n$.*

**Theorem 6.** *The approximate Bahadur slope of the linear-time kernel Stein (LKS) test statistic $\sqrt{n}\widehat{S_l^2}$ is $c^{(\text{LKS})} = \frac{1}{2} \frac{\left[\mathbb{E}_q h_p(\mathbf{x}, \mathbf{x}')\right]^2}{\mathbb{E}_p\left[h_p^2(\mathbf{x}, \mathbf{x}')\right]}$, where $h_p$ is the U-statistic kernel of the KSD statistic, and $\rho(n) = n$.*

To make these results concrete, we consider the setting where $p = \mathcal{N}(0, 1)$ and $q = \mathcal{N}(\mu_q, 1)$. We assume that both tests use the Gaussian kernel $k(x, y) = \exp\left(-(x - y)^2/2\sigma_k^2\right)$, possibly with different bandwidths. We write $\sigma_k^2$ and $\kappa^2$ for the FSSD and LKS bandwidths, respectively. Under these assumptions, the slopes given in Theorem 5 and Theorem 6 can be derived explicitly. The full expressions of the slopes are given in Proposition 12 and Proposition 13 (in the appendix). By [12, 13] (recalled as Theorem 10 in the supplement), the approximate Bahadur efficiency can be computed by taking the ratio of the two slopes. The efficiency is given in Theorem 7.

**Theorem 7** (Efficiency in the Gaussian mean shift problem). *Let $E_1(\mu_q, v, \sigma_k^2, \kappa^2)$ be the approximate Bahadur efficiency of $n\widehat{\text{FSSD}^2}$ relative to $\sqrt{n}\widehat{S_l^2}$ for the case where $p = \mathcal{N}(0, 1), q = \mathcal{N}(\mu_q, 1)$, and $J = 1$ (i.e., one test location $v$ for $n\widehat{\text{FSSD}^2}$). Fix $\sigma_k^2 = 1$ for $n\widehat{\text{FSSD}^2}$. Then, for any $\mu_q \neq 0$, for some $v \in \mathbb{R}$, and for any $\kappa^2 > 0$, we have $E_1(\mu_q, v, \sigma_k^2, \kappa^2) > 2$.*

When $p = \mathcal{N}(0, 1)$ and $q = \mathcal{N}(\mu_q, 1)$ for $\mu_q \neq 0$, Theorem 7 guarantees that our FSSD test is asymptotically at least twice as efficient as the LKS test in the Bahadur sense. We note that the

efficiency is conservative in the sense that $\sigma_k^2 = 1$ regardless of $\mu_q$. Choosing $\sigma_k^2$ dependent on $\mu_q$ will likely improve the efficiency further.

# 5   Experiments

In this section, we demonstrate the performance of the proposed test on a number of problems. The primary goal is to understand the conditions under which the test can perform well.

**Sensitivity to Local Differences**   We start by demonstrating that the test power objective $\mathrm{FSSD}^2/\sigma_{H_1}$ captures local differences of $p$ and $q$, and that interpretable features $v$ are found. Consider a one-dimensional problem in which $p = \mathcal{N}(0,1)$ and $q = \mathrm{Laplace}(0, 1/\sqrt{2})$, a zero-mean Laplace distribution with scale parameter $1/\sqrt{2}$. These parameters are chosen so that $p$ and $q$ have the same mean and variance. Figure 1 plots the (rescaled) objective as a function of $v$. The objective illustrates that the best features (indicated by $v^*$) are at the most discriminative locations.

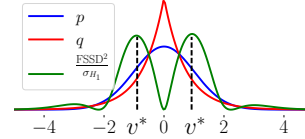

Figure 1: The power criterion $\mathrm{FSSD}^2/\sigma_{H_1}$ as a function of test location $v$.

**Test Power**  We next investigate the power of different tests on two problems:

1. **Gaussian vs. Laplace**: $p(\mathbf{x}) = \mathcal{N}(\mathbf{x}|\mathbf{0}, \mathbf{I}_d)$ and $q(\mathbf{x}) = \prod_{i=1}^{d} \mathrm{Laplace}(x_i|0, 1/\sqrt{2})$ where the dimension $d$ will be varied. The two distributions have the same mean and variance. The main characteristic of this problem is local differences of $p$ and $q$ (see Figure 1). Set $n = 1000$.

2. **Restricted Boltzmann Machine** (RBM): $p(\mathbf{x})$ is the marginal distribution of $p(\mathbf{x}, \mathbf{h}) = \frac{1}{Z}\exp\left(\mathbf{x}^\top \mathbf{B}\mathbf{h} + \mathbf{b}^\top\mathbf{x} + \mathbf{c}^\top\mathbf{x} - \frac{1}{2}\|\mathbf{x}\|^2\right)$, where $\mathbf{x} \in \mathbb{R}^d$, $\mathbf{h} \in \{\pm 1\}^{d_h}$ is a random vector of hidden variables, and $Z$ is the normalization constant. The exact marginal density $p(\mathbf{x}) = \sum_{\mathbf{h}\in\{-1,1\}^{d_h}} p(\mathbf{x}, \mathbf{h})$ is intractable when $d_h$ is large, since it involves summing over $2^{d_h}$ terms. Recall that the proposed test only requires the score function $\nabla_{\mathbf{x}} \log p(\mathbf{x})$ (not the normalization constant), which can be computed in closed form in this case. In this problem, $q$ is another RBM where entries of the matrix $\mathbf{B}$ are corrupted by Gaussian noise. This was the problem considered in [22]. We set $d = 50$ and $d_h = 40$, and generate samples by $n$ independent chains (i.e., $n$ independent samples) of blocked Gibbs sampling with 2000 burn-in iterations.

We evaluate the following six kernel-based nonparametric tests with $\alpha = 0.05$, all using the Gaussian kernel. **1. FSSD-rand**: the proposed FSSD test where the test locations set to random draws from a multivariate normal distribution fitted to the data. The kernel bandwidth is set by the commonly used median heuristic i.e., $\sigma_k = \mathrm{median}(\{\|\mathbf{x}_i - \mathbf{x}_j\|, i < j\})$. **2. FSSD-opt**: the proposed FSSD test where both the test locations and the Gaussian bandwidth are optimized (Section 3.2). **3. KSD**: the quadratic-time Kernel Stein Discrepancy test with the median heuristic. **4. LKS**: the linear-time version of KSD with the median heuristic. **5. MMD-opt**: the quadratic-time MMD two-sample test of [16] where the kernel bandwidth is optimized by grid search to maximize a power criterion as described in [29]. **6. ME-opt**: the linear-time mean embeddings (ME) two-sample test of [19] where parameters are optimized. We draw $n$ samples from $p$ to run the two-sample tests (MMD-opt, ME-opt). For FSSD tests, we use $J = 5$ (see Section A for an investigation of test power as $J$ varies). All tests with optimization use 20% of the sample size $n$ for parameter tuning. Code is available at https://github.com/wittawatj/kernel-gof.

Figure 2 shows the rejection rates of the six tests for the two problems, where each problem is repeated for 200 trials, resampling $n$ points from $q$ every time. In Figure 2a (Gaussian vs. Laplace), high performance of FSSD-opt indicates that the test performs well when there are local differences between $p$ and $q$. Low performance of FSSD-rand emphasizes the importance of the optimization of FSSD-opt to pinpoint regions where $p$ and $q$ differ. The power of KSD quickly drops as the dimension increases, which can be understood since KSD is the RKHS norm of a function witnessing differences in $p$ and $q$ across the entire domain, including where these differences are small.

We next consider the case of RBMs. Following [22], $\mathbf{b}, \mathbf{c}$ are independently drawn from the standard multivariate normal distribution, and entries of $\mathbf{B} \in \mathbb{R}^{50\times40}$ are drawn with equal probability from $\{\pm1\}$, in each trial. The density $q$ represents another RBM having the same $\mathbf{b}, \mathbf{c}$ as in $p$, and with all entries of $\mathbf{B}$ corrupted by independent zero-mean Gaussian noise with standard deviation $\sigma_{per}$. Figure

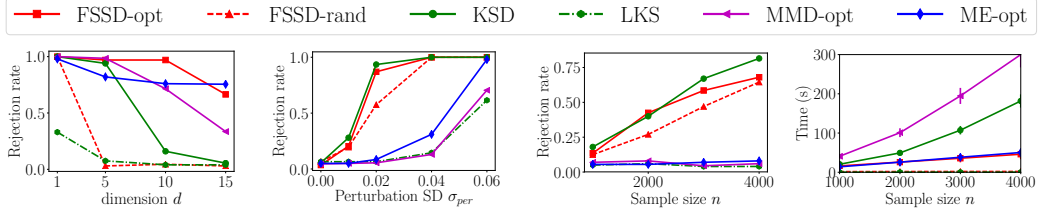

(a) Gaussian vs. Laplace. $n = 1000$.

(b) RBM. $n = 1000$. Perturb all entries of $\mathbf{B}$.

(c) RBM. $\sigma_{per} = 0.1$. Perturb $B_{1,1}$.

(d) Runtime (RBM)

Figure 2: Rejection rates of the six tests. The proposed linear-time FSSD-opt has a comparable or higher test power in some cases than the quadratic-time KSD test.

2b shows the test powers as $\sigma_{per}$ increases, for a fixed sample size $n = 1000$. We observe that all the tests have correct false positive rates (type-I errors) at roughly $\alpha = 0.05$ when there is no perturbation noise. In particular, the optimization in FSSD-opt does not increase false positive rate when $H_0$ holds. We see that the performance of the proposed FSSD-opt matches that of the quadratic-time KSD at all noise levels. MMD-opt and ME-opt perform far worse than the goodness-of-fit tests when the difference in $p$ and $q$ is small ($\sigma_{per}$ is low), since these tests simply represent $p$ using samples, and do not take advantage of its structure.

The advantage of having $\mathcal{O}(n)$ runtime can be clearly seen when the problem is much harder, requiring larger sample sizes to tackle. Consider a similar problem on RBMs in which the parameter $\mathbf{B} \in \mathbb{R}^{50 \times 40}$ in $q$ is given by that of $p$, where only the first entry $B_{1,1}$ is perturbed by random $\mathcal{N}(0, 0.1^2)$ noise. The results are shown in Figure 2c where the sample size $n$ is varied. We observe that the two two-sample tests fail to detect this subtle difference even with large sample size. The test powers of KSD and FSSD-opt are comparable when $n$ is relatively small. It appears that KSD has higher test power than FSSD-opt in this case for large $n$. However, this moderate gain in the test power comes with an order of magnitude more computation. As shown in Figure 2d, the runtime of the KSD is much larger than that of FSSD-opt, especially at large $n$. In these problems, the performance of the new test (even without optimization) far exceeds that of the LKS test. Further simulation results can be found in Section B.

**Interpretable Features** In the final simulation, we demonstrate that the learned test locations are informative in visualising where the model does not fit the data well. We consider crime data from the Chicago Police Department, recording $n = 11957$ locations (latitude-longitude coordinates) of robbery events in Chicago in 2016.[3] We address the situation in which a model $p$ for the robbery location density is given, and we wish to visualise where it fails to match the data.

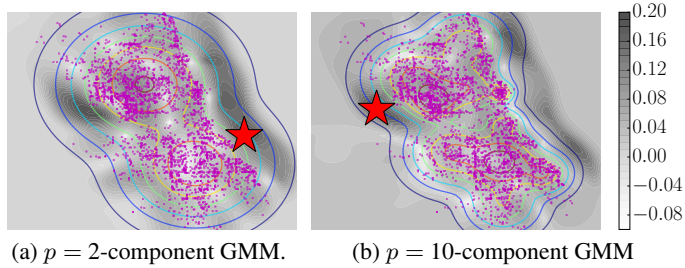

(a) $p = 2$-component GMM.

(b) $p = 10$-component GMM

Figure 3: Plots of the optimization objective as a function of test location $\mathbf{v} \in \mathbb{R}^2$ in the Gaussian mixture model (GMM) evaluation task.

We fit a Gaussian mixture model (GMM) with the expectation-maximization algorithm to a subsample of 5500 points. We then test the model on a held-out test set of the same size to obtain proposed locations of relevant features $\mathbf{v}$. Figure 3a shows the test robbery locations in purple, the model with two Gaussian components in wireframe, and the optimization objective for $\mathbf{v}$ as a grayscale contour plot (a red star indicates the maximum). We observe that the 2-component model is a poor fit to the data, particularly in the right tail areas of the data, as indicated in dark gray (i.e., the objective is high). Figure 3b shows a similar plot with a 10-component GMM. The additional components appear to have eliminated some mismatch in the right tail, however a discrepancy still exists in the left region. Here, the data have a sharp boundary on the right side following the geography of Chicago, and do not exhibit exponentially decaying Gaussian-like tails. We note that tests based on a learned feature located at the maximum both correctly reject $H_0$.

**Acknowledgement**

WJ, WX, and AG thank the Gatsby Charitable Foundation for the financial support. ZSz was financially supported by the Data Science Initiative. KF has been supported by KAKENHI Innovative Areas 25120012.

## Footnotes

[3]Data can be found at `https://data.cityofchicago.org`.

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
