[Supplementary Material · kgof_nips2017_sup.pdf]

# A Linear-Time Kernel Goodness-of-Fit Test

# Supplementary

## A Rejection Rate vs. Number of Test Locations $J$

(a) SG. $d = 5$. $\alpha = 0.05$.  (b) Gaussian vs. GMM. $d = 1$.  (c) GVD. $d = 5$.

Figure 4: Plots of rejection rate against the number of test locations $J$ in the three toy problems in Section A.

The aim of this section is to explore the test power of the proposed FSSD test as a function of the number of test locations $J$. We consider three synthetic problems to illustrate three phenomena depending on the characteristic of the problem. We note that the test power may not necessarily increase with $J$. Figure 4 shows the rejection rate as a function of the test locations $J$ in the three problems described below. In all cases, the sample size is set to $n = 500$, the train/test ratio is 50%, and the significance level is $\alpha = 0.05$. All rejection rates are computed with 200 trials with data sampled from the specified $q$ in every trial.

We emphasize that the FSSD test is not designed to be used with large $J$, since doing so defeats the purpose of a linear-time test. We show in the main text in Section 2 that using $J = 5$ is typically sufficient in practice.

**Same Gaussian (SG):**     In this problem, $p = q = \mathcal{N}(\mathbf{0}, \mathbf{I})$ in $\mathbb{R}^5$ i.e., $H_0$ is true. It can be seen in Figure 4a that both the FSSD tests with and without optimization achieve correct false positive rate at roughly $\alpha$ for all $J$ considered. That is, under $H_0$, the false rejection rate stays at the right level for all $J$.

**Gaussian vs. Gaussian mixture model (GMM):**     This is a one-dimensional problem where $p = \mathcal{N}(0, 1)$ and $q = 0.9\mathcal{N}(0, 1) + 0.1\mathcal{N}(0, 0.1^2)$ i.e., a mixture of two normal distributions. In this problem, $p$ significantly differs from $q$ in a small region around 0. This difference is created by the second mixture component. The characteristic of this problem is the local difference of $p$ and $q$.

Figure 4b indicates that using random test locations (FSSD-rand) does not give high test power. With optimization (FSSD-opt), the power increases as $J$ increases up to a point, after which it slightly drops down and reaches a plateau. This behavior can be explained by noting that there is only a very small region around 0 to detect the difference. More signal can be gained with diminishing return by increasing the number of test locations around 0. When $J$ is sufficiently high, the increase in the variance of the statistic outweighs the gain of the signal (recall that the variance of the null distribution increases with $J$). This increase in the variance reduces the test power.

**Gaussian Variance Difference (GVD):**     This is a synthetic problem studied in [19] where $p = \mathcal{N}(\mathbf{0}, \mathbf{I})$ and $q = \mathcal{N}(\mathbf{0}, \mathrm{diag}(2, 1 \ldots, 1))$ in $\mathbb{R}^5$. In this case, the region of difference between $q$ and $p$ exists only along the first dimension, and is broad.

In this case, Figure 4c shows that, with optimization, the power increases as the number of test locations increases. Unlike the case of Gaussian vs. GMM, the region of difference in this case is broad, and can accommodate more test locations to increase the signal. Despite this, we expect the test power to reach a plateau when $J$ is sufficiently large for the same reason as described previously. In FSSD-rand, random test locations decrease the power due to the increase in the variance. Since only one dimension is relevant in determining the difference of $p$ and $q$, it is unlikely that random locations are in the right region.

# B More Experiments

(a) RBM. $n = 1000$. Perturb all entries of $\mathbf{B}$.

(b) RBM. $\sigma_{per} = 0.1$. Perturb $B_{1,1}$.

(c) Runtime (RBM)

(d) RBM. No perturbation. $H_0$ holds.

Figure 5: Rejection rates of the six tests in the RBM problem with $d = 50$ and $d_h = 10$.

(a) $d = 50, d_h = 10$

(b) $d = 50, d_h = 40$

Figure 6: Pairwise scatter plots of 1000 points drawn from RBMs. Only the first 4 variates out of 50 are shown. **(a)**: RBM with $d = 50$ dimensions with $d_h = 10$ latent variables. **(b)**: RBM with $d = 50$ dimensions with $d_h = 40$ latent variables.

Recall that in Section 5, we evaluate the test powers of all the six tests on the RBM problem with $d = 50$ and $d_h = 40$ (i.e., the number of latent variables). We aim to provide more evaluations in this section. In [22], the setting of $d = 50$ and $d_h = 10$ was studied. Here we consider the same setting and show the results in Figure 5 where all other problem configurations are the same as in Section 5.

In Figure 5a, $p$ is set to an RBM with parameters randomly drawn (described in Section 5), and $q$ is the same RBM with all entries of the parameter $\mathbf{B} \in \mathbb{R}^{50 \times 10}$ perturbed by independent Gaussian noise with standard deviation $\sigma_{per}$, which varies from 0 to 0.06. We observe that the proposed FSSD-opt and KSD perform comparably. Figure 5b considers a hard problem where only the first entry $B_{1,1}$ is perturbed by noise following $\mathcal{N}(0, 0.1^2)$, and the sample size $n$ is varied. In both of these two cases, the overall trend is similar to the case of $d = 50$ and $d_h = 40$ presented in Figure 2. It is interesting to note that FSSD-rand, relying on random test locations, performs comparably or even outperforms FSSD-opt in the case of $d = 50, d_h = 10$, but not in the case of $d = 50, d_h = 40$. This phenomenon can be explained as follows. In the case of $d = 50, d_h = 10$, the data generated from the RBM tend to have simple structure (see Figure 6a). By contrast, data generated from the RBM with $d = 50, d_h = 40$ (more latent variables) have larger variance, and can form a complicated structure (Figure 6b), requiring a careful choice of test locations to detect differences of $p$ and $q$. When $d = 50, d_h = 10$, however, random test locations given by random draws from a Gaussian distribution fitted to the data are sufficient to capture the simple structural difference. This explains why FSSD-rand can perform well in this case. Additionally, FSSD-rand also has 20% more testing data, since FSSD-opt uses 20% of the sample for parameter tuning.

Figure 5d shows the rejection rates of all the tests as the sample size increases when $p$ and $q$ are the same RBM. All the tests have roughly the right false rejection rates at the set significance level $\alpha = 0.05$.

## C Proof of Theorem 1

Recall Theorem 1:

**Theorem 1** (The Finite Set Stein Discrepancy (FSSD))**.** *Let* $V = \{\mathbf{v}_1, \ldots, \mathbf{v}_J\} \subset \mathbb{R}^d$ *be random vectors drawn i.i.d. from a distribution* $\eta$ *which has a density. Let* $\mathcal{X}$ *be a connected open set in* $\mathbb{R}^d$. *Define* $\mathrm{FSSD}_p^2(q) := \frac{1}{dJ} \sum_{i=1}^d \sum_{j=1}^J g_i^2(\mathbf{v}_j)$. *Assume that 1)* $k \colon \mathcal{X} \times \mathcal{X} \to \mathbb{R}$ *is* $C_0$-*universal [6, Definition 4.1] and real analytic i.e., for all* $\mathbf{v} \in \mathcal{X}$, $f(\mathbf{x}) := k(\mathbf{x}, \mathbf{v})$ *is a real analytic function on* $\mathcal{X}$. *2)* $\mathbb{E}_{\mathbf{x} \sim q} \mathbb{E}_{\mathbf{x}' \sim q} h_p(\mathbf{x}, \mathbf{x}') < \infty$. *3)* $\mathbb{E}_{\mathbf{x} \sim q} \|\nabla_{\mathbf{x}} \log p(\mathbf{x}) - \nabla_{\mathbf{x}} \log q(\mathbf{x})\|^2 < \infty$. *4)* $\lim_{\|\mathbf{x}\| \to \infty} p(\mathbf{x}) \mathbf{g}(\mathbf{x}) = 0$.

*Then, for any* $J \geq 1$, $\eta$-*almost surely* $\mathrm{FSSD}_p^2(q) = 0$ *if and only if* $p = q$.

*Proof.* Since $k$ is real analytic, the components $g_1, \ldots, g_d$ of $\mathbf{g}$ are real analytic by Lemma 15. For each $i = 1, \ldots, d$, if $g_i$ is real analytic, then $\sum_{j=1}^J g_i^2(\mathbf{v}_j) = 0$ if and only if $g_i(\mathbf{y}) = 0$ for all $\mathbf{y} \in \mathcal{X}$, $\eta$-almost surely (require that the domain $\mathcal{X}$ be a connected open set) [24]. This implies that $\frac{1}{dJ} \sum_{i=1}^d \sum_{j=1}^J g_i^2(\mathbf{v}_j) = 0$ if and only if $\mathbf{g}(\mathbf{y}) = \mathbf{0}$ for all $\mathbf{y} \in \mathcal{X}$, $\eta$-almost surely. By Theorem 14, $\mathbf{g} = \mathbf{0}$ (the zero function) if and only if $p = q$. $\square$

## D More on Bahadur Slope

In practice, the main difficulty in determining the approximate Bahadur slope is the computation of $-2 \operatorname{plim}_{n \to \infty} \frac{\log(1 - F(T_n))}{\rho(n)}$, typically requiring the aid of the theory of large deviations. There are further sufficient conditions which make the computation easier. The following conditions are due to [12, 13], first appearing in [1] in a slightly less general form.

**Definition 8.** Let $\mathcal{D}(a, t)$ be a class of all continuous cumulative distribution functions (CDF) $F$ such that $-2 \log(1 - F(x)) = ax^t(1 + o(1))$, as $x \to \infty$ for $a > 0$ and $t > 0$.

**Theorem 9** ([12, 13])**.** *Consider a sequence of test statistic* $T_n$. *Assume that*

1. *There exists a function* $F(x)$ *such that for* $\theta \in \Theta_0$, $\lim_{n \to \infty} P_\theta(T_n < x) = F(x)$, *for all* $x$, *and such that* $F \in \mathcal{D}(a, t)$ *for some* $a > 0$ *and* $t > 0$ *(see Definition 8).*

2. *There exists a continuous, strictly increasing function* $R : (0, \infty) \to (0, \infty)$ *with* $\lim_{n \to \infty} R(n) = \infty$, *and a function* $b(\theta)$ *with* $0 < b(\theta) < \infty$ *defined on* $\Theta \backslash \Theta_0$, *such that for all* $\theta \in \Theta \backslash \Theta_0$, $\operatorname{plim}_{n \to \infty} T_n / R(n) = b(\theta)$.

*Then,* $-2 \operatorname{plim}_{n \to \infty} \frac{\log(1 - F(T_n))}{[R(n)]^t} = a [b(\theta)]^t =: c(\theta)$, *the approximate slope of the sequence* $T_n$, *where* $\rho(n) = R(n)^t$ *(see Section 4).*

**Theorem 10** ([12, 13])**.** *Consider two sequences of test statistics* $T_n^{(1)}$ *and* $T_n^{(2)}$. *Let* $F^{(i)}$ *be the CDF of* $T_n^{(i)}$ *for* $i = 1, 2$. *Assume that each sequence satisfies all the conditions in Theorem 9 with* $F^{(i)} \in \mathcal{D}(a_i, t_i)$. *Further, assume that* $\left[R^{(1)}(x)\right]^{t_1} = \left[R^{(2)}(x)\right]^{t_2}$ *for all* $x$. *Then*

$$\operatorname*{plim}_{n \to \infty} \frac{\log(1 - F^{(1)}(T_n^{(1)}))}{\log(1 - F^{(2)}(T_n^{(2)}))} = \frac{c^{(1)}(\theta)}{c^{(2)}(\theta)} = \varphi_{1,2}(\theta),$$

*which is the approximate Bahadur efficiency of* $T_n^{(1)}$ *relative to* $T_n^{(2)}$.

With Theorem 9, the difficulty is in showing that $F \in \mathcal{D}(a, t)$ for some $a > 0, t > 0$. Typically verification of the assumption 2 of Theorem 9 poses no problem. [1] showed that the CDF of $\mathcal{N}(0, 1)$ belongs to $\mathcal{D}(1, 2)$ and the CDF of $\chi_k^2$ (chi-squared distribution with $k$ degrees of freedom, fixed $k$) belongs to $\mathcal{D}(1, 1)$. The following results make it easier to determine whether a given CDF is in the class $\mathcal{D}(a, t)$.

**Theorem 11** ([13, Theorem 6, 7])**.** *Let* $X$ *have CDF* $F \in \mathcal{D}(a, t)$, *and* $X_1, \ldots, X_m$ *be independent random variables, each with CDF* $F_i \in \mathcal{D}(a, t)$. *Then, the following statements are true.*

1. *If* $b > 0$, *then the CDF of* $bX$ *is in* $\mathcal{D}(ab^{-t}, t)$.

2. $X - b$ has CDF in $\mathcal{D}(a, t)$ provided that $t \geq 1$.

3. For $r > 0$, $X^r$ has CDF in $\mathcal{D}(a, r^{-1}t)$ provided that $F(0) = 0$.

4. $\max(X_1, \ldots, X_m)$ has CDF in $\mathcal{D}(a, t)$.

5. Let $a_1, \ldots, a_m$ be non-negative real numbers such that $a_{max} := \max(a_1, \ldots, a_m) > 0$. Then, $\sum_{i=1}^m a_i X_i$ has CDF in $\mathcal{D}(a \cdot a_{max}^{-t}, t)$ provided that $\sum_{i=1}^m X_i$ has CDF in $\mathcal{D}(a, t)$ and $X_i \geq 0$ for all $i = 1, \ldots, m$.

## E  Proof of Theorem 3

Recall Theorem 3:

**Theorem 3.** Let $\hat{\boldsymbol{\Sigma}}_q := \frac{1}{n} \sum_{i=1}^n \boldsymbol{\tau}(\mathbf{x}_i) \boldsymbol{\tau}^\top(\mathbf{x}_i) - [\frac{1}{n} \sum_{i=1}^n \boldsymbol{\tau}(\mathbf{x}_i)][\frac{1}{n} \sum_{j=1}^n \boldsymbol{\tau}(\mathbf{x}_j)]^\top$ with $\{\mathbf{x}_i\}_{i=1}^n \sim q$. Suppose that the test threshold $T_\alpha$ is set to the $(1-\alpha)$-quantile of the distribution of $\sum_{i=1}^{dJ}(Z_i^2 - 1)\hat{\nu}_i$ where $\{Z_i\}_{i=1}^{dJ} \overset{i.i.d.}{\sim} \mathcal{N}(0,1)$, and $\hat{\nu}_1, \ldots, \hat{\nu}_{dJ}$ are eigenvalues of $\hat{\boldsymbol{\Sigma}}_q$. Then, under $H_0$, asymptotically the false positive rate is $\alpha$. Under $H_1$, for $\{\mathbf{v}_j\}_{j=1}^J$ drawn from a distribution with a density, the test power $\mathbb{P}_{H_1}(n\widehat{\mathrm{FSSD}^2} > T_\alpha) \to 1$ as $n \to \infty$.

*Proof.* Under $H_0$, $p = q$ implies that $\hat{\boldsymbol{\Sigma}}_q = \hat{\boldsymbol{\Sigma}}_p$ (empirical estimate of $\boldsymbol{\Sigma}_p$). Let $\lambda_j(A)$ denote the $j^{th}$ eigenvalue of the matrix $A$. Lemma 16 implies that $A \mapsto \lambda_j(A)$ is continuous on the space of real symmetric matrices, for all $j$. Since $\mathrm{plim}_{n\to\infty} \|\hat{\boldsymbol{\Sigma}}_p - \boldsymbol{\Sigma}_p\| = 0$, by the continuous mapping theorem, the eigenvalues of $\hat{\boldsymbol{\Sigma}}_p$ converge to the eigenvalues of $\boldsymbol{\Sigma}_p$ in probability. This implies that $\sum_{i=1}^{dJ}(Z_i^2 - 1)\hat{\nu}_i$ converges in probability to $\sum_{i=1}^{dJ}(Z_i^2 - 1)\omega_i$ as $n \to \infty$, where $\{\omega_i\}_{i=1}^{dJ}$ are eigenvalues of $\boldsymbol{\Sigma}_p$. By Lemma 17, the quantile also converges, and the test threshold thus matches that of the true asymptotic null distribution given in claim 1 of Proposition 2.

Assume $H_1$ holds. Let $\hat{t}_\alpha, t_\alpha$ be $(1 - \alpha)$-quantiles of the distributions of $\sum_{i=1}^{dJ}(Z_i^2 - 1)\hat{\nu}_i$ and $\sum_{i=1}^{dJ}(Z_i^2 - 1)\nu_i$, respectively, where $\{\nu_i\}_{i=1}^{dJ}$ are eigenvalues of $\boldsymbol{\Sigma}_q$. By the same argument as in the previous paragraph, $\hat{t}_\alpha$ converges in probability to $t_\alpha$, which is a constant independent of the sample size $n$. Given $\{\mathbf{v}_j\}_{j=1}^J \sim \eta$, where $\eta$ is a distribution with a density, $\mathrm{FSSD}^2 > 0$ by Theorem 1. It follows that

$$\lim_{n\to\infty} \mathbb{P}\left(n\widehat{\mathrm{FSSD}^2} > \hat{t}_\alpha\right) = \lim_{n\to\infty} \mathbb{P}\left(\widehat{\mathrm{FSSD}^2} - \frac{\hat{t}_\alpha}{n} > 0\right) \overset{(a)}{=} \mathbb{P}\left(\mathrm{FSSD}^2 > 0\right) = 1,$$

where at $(a)$, we use the fact that $\widehat{\mathrm{FSSD}^2}$ converges in probability to $\mathrm{FSSD}^2$ by the law of large numbers, and that $\lim_{n\to\infty} \hat{t}_\alpha/n = 0$. $\qquad\square$

## F  Proof of Theorem 5 (Slope of $n\widehat{\mathrm{FSSD}^2}$)

Recall Theorem 5:

**Theorem 5.** The approximate Bahadur slope of $n\widehat{\mathrm{FSSD}^2}$ is $c^{(\mathrm{FSSD})} := \mathrm{FSSD}^2/\omega_1$, where $\omega_1$ is the maximum eigenvalue of $\boldsymbol{\Sigma}_p := \mathbb{E}_{\mathbf{x}\sim p}[\boldsymbol{\tau}(\mathbf{x})\boldsymbol{\tau}^\top(\mathbf{x})]$ and $\rho(n) = n$.

*Proof.* We will use Theorem 9 to derive the slope. For the assumption 1 of Theorem 9, we first show that the asymptotic null distribution belongs to the class $\mathcal{D}(a = 1/\omega_1, t = 1)$ as defined in Definition 8. By Proposition 2, the asymptotic null distribution is $\sum_{i=1}^{dJ} \omega_i Z_i^2 - \sum_{i=1}^{dJ} \omega_i$ where $Z_1, \ldots, Z_{dJ} \overset{i.i.d.}{\sim} \mathcal{N}(0,1)$ and $\omega_1 \geq \cdots \geq \omega_{dJ} \geq 0$ are eigenvalues of $\boldsymbol{\Sigma}_p$. It is known from [1] that the CDF of $\chi_f^2$ is in $\mathcal{D}(1, 1)$ for any fixed degrees of freedom $f$. Thus, it follows from claim 5 of Theorem 11 that the CDF of $\sum_{i=1}^{dJ} \omega_i Z_i^2$ is in $\mathcal{D}(a = 1/\omega_1, t = 1)$. Claim 2 of Theorem 11 guarantees that the CDF of $\sum_{i=1}^{dJ} \omega_i Z_i^2 - \sum_{i=1}^{dJ} \omega_i$ is in $\mathcal{D}(a = 1/\omega_1, t = 1)$ as desired.

For assumption 2 of Theorem 9, choose $R(n) := n$. It follows from the weak law of large numbers that under $H_1$, $n\widehat{\mathrm{FSSD}^2}/R(n) \overset{p}{\to} \mathrm{FSSD}^2$. By Theorem 9, the approximate slope is $\mathrm{FSSD}^2/\omega_1$. $\qquad\square$

# G Proof of Theorem 6 (Slope of $\sqrt{n}\widehat{S_l^2}$)

Recall Theorem 6:

**Theorem 6.** *The approximate Bahadur slope of the linear-time kernel Stein (LKS) test statistic $\sqrt{n}\widehat{S_l^2}$ is $c^{(\text{LKS})} = \frac{1}{2}\frac{\left[\mathbb{E}_q h_p(\mathbf{x},\mathbf{x}')\right]^2}{\mathbb{E}_p\left[h_p^2(\mathbf{x},\mathbf{x}')\right]}$, where $h_p$ is the U-statistic kernel of the KSD statistic, and $\rho(n) = n$.*

*Proof.* We will use Theorem 9 to derive the slope. By the central limit theorem,

$$\sqrt{n}\left(\widehat{S_l^2} - S_p^2(q)\right) \xrightarrow{d} \mathcal{N}(0, 2\mathbb{V}_q[h_p(\mathbf{x},\mathbf{x}')]),$$

where $\mathbb{V}_q[h_p(\mathbf{x},\mathbf{x}')] := \mathbb{E}_{\mathbf{x}\sim q}\mathbb{E}_{\mathbf{x}'\sim q}[h_p^2(\mathbf{x},\mathbf{x}')] - (\mathbb{E}_{\mathbf{x}\sim q}\mathbb{E}_{\mathbf{x}'\sim q}[h_p(\mathbf{x},\mathbf{x}')])^2$. Under $H_0 : p = q$, it follows that $S_p^2(q) = \mathbb{E}_{\mathbf{x}\sim q}\mathbb{E}_{\mathbf{x}'\sim q}[h_p(\mathbf{x},\mathbf{x}')] = 0$ by Theorem 14, and $\sqrt{n}\widehat{S_l^2} \xrightarrow{d} \mathcal{N}(0, 2\mathbb{V}_p[h_p(\mathbf{x},\mathbf{x}')])$ where $\mathbb{V}_p[h_p(\mathbf{x},\mathbf{x}')] := \mathbb{E}_{\mathbf{x}\sim p}\mathbb{E}_{\mathbf{x}'\sim p}[h_p^2(\mathbf{x},\mathbf{x}')]$. It is known from [1] that the CDF of $\mathcal{N}(0,1)$ is in the class $\mathcal{D}(1,2)$ (see Definition 8). Thus, by property 1 of Theorem 11, the CDF of $\mathcal{N}(0, 2\mathbb{V}_p[h_p(\mathbf{x},\mathbf{x}')])$ is in $\mathcal{D}\left(a = \frac{1}{2\mathbb{V}_p[h_p(\mathbf{x},\mathbf{x}')]}, t = 2\right)$.

For assumption 2 of Theorem 9, choose $R(n) := \sqrt{n}$. It follows from the weak law of large numbers that under $H_1$, $\sqrt{n}\widehat{S_l^2}/R(n) = \widehat{S_l^2} \xrightarrow{p} S_p^2(q)$. By Theorem 9, the approximate slope is $\frac{S_p^4(q)}{2\mathbb{V}_p[h_p(\mathbf{x},\mathbf{x}')]}$. $\qquad\square$

# H Proof of Theorem 7

We will first prove a number of useful results that will allow us to prove Theorem 7 at the end. Recall that $v$ denotes a test location in the FSSD test, $\sigma_k^2$ denotes the Gaussian kernel bandwidth of the FSSD test, and $\kappa^2$ denotes the Gaussian kernel bandwidth of the LKS test.

**Proposition 12.** *Under the assumption that $J = 1$ (i.e., one test location $v$), $p = \mathcal{N}(0,1)$ and $q = \mathcal{N}(\mu_q, \sigma_q^2)$, the approximate Bahadur Slope of $n\widehat{\text{FSSD}^2}$ is*

$$c^{(\text{FSSD})} := \frac{\left(\sigma_k^2\right)^{3/2}\left(\sigma_k^2 + 2\right)^{5/2} e^{\frac{v^2}{\sigma_k^2+2} - \frac{(v-\mu_q)^2}{\sigma_k^2+\sigma_q^2}}\left(\left(\sigma_k^2 + 1\right)\mu_q + v\left(\sigma_q^2 - 1\right)\right)^2}{\left(\sigma_k^2 + \sigma_q^2\right)^3\left(\sigma_k^6 + 4\sigma_k^4 + (v^2 + 5)\sigma_k^2 + 2\right)}. \tag{3}$$

*Proof.* This result follows directly from Theorem 5 specialized to the case of $p = \mathcal{N}(0,1)$, $q = \mathcal{N}(\mu_q, \sigma_q^2)$, and $J = 1$. Since $dJ = 1$, the covariance matrix

$$\boldsymbol{\Sigma}_p = \mathbb{E}_{x\sim p}\left[\xi_p^2(x,v)\right] = \frac{e^{-\frac{v^2}{\sigma_k^2+2}}\left(\sigma_k^6 + 4\sigma_k^4 + \left(v^2 + 5\right)\sigma_k^2 + 2\right)}{\sigma_k\left(\sigma_k^2 + 2\right)^{5/2}}$$

reduces to a scalar, where $\xi_p(x,v) = \left[\frac{\partial}{\partial x}\log p(x)\right]k(x,v) + \frac{\partial}{\partial x}k(x,v) = -e^{-\frac{(v-x)^2}{2\sigma_k^2}}\left(x\sigma_k^2 - v + x\right)/\sigma_k^2$. In this case,

$$\text{FSSD}^2 = \mathbb{E}_{x\sim q}^2\left[\xi_p(x,v)\right] = \frac{\sigma_k^2 e^{-\frac{(v-\mu_q)^2}{\sigma_k^2+\sigma_q^2}}\left(\left(\sigma_k^2 + 1\right)\mu_q + v\left(\sigma_q^2 - 1\right)\right)^2}{\left(\sigma_k^2 + \sigma_q^2\right)^3}.$$

Taking the ratio $\text{FSSD}^2/\mathbb{E}_{x\sim p}\left[\xi_p^2(x,v)\right]$ gives the result. $\qquad\square$

**Proposition 13.** *Assume that $p = \mathcal{N}(0,1)$ and $q = \mathcal{N}(\mu_q, \sigma_q^2)$. Let $\sqrt{n}\widehat{S_l^2}$ be the linear-time kernel Stein (LKS) test statistic where $\widehat{S_l^2}$ is defined in Section 2 with a Gaussian kernel $k(x,y) = \exp\left(-\frac{(x-y)^2}{2\kappa^2}\right)$. Then, the following statements hold.*

*1. The population kernel Stein discrepancy is*

$$S_p^2(q) = \frac{\mu_q^2\left(\kappa^2 + 2\sigma_q^2\right) + \left(\sigma_q^2 - 1\right)^2}{\left(\kappa^2 + 2\sigma_q^2\right)\sqrt{\frac{2\sigma_q^2}{\kappa^2} + 1}}.$$

*2. The approximate Bahadur slope of $\sqrt{n}\widehat{S_l^2}$ is*

$$c^{(\mathrm{LKS})} := \frac{\kappa^5\left(\kappa^2 + 4\right)^{5/2}\left[\mu_q^2\left(\kappa^2 + 2\sigma_q^2\right) + \left(\sigma_q^2 - 1\right)^2\right]^2}{2\left(\kappa^8 + 8\kappa^6 + 21\kappa^4 + 20\kappa^2 + 12\right)\left(\kappa^2 + 2\sigma_q^2\right)^3}. \tag{4}$$

*3. Let*

$$c_1^{(\mathrm{LKS})} = \frac{\left(\kappa^2\right)^{5/2}\left(\kappa^2 + 4\right)^{5/2}\mu_q^4}{2\left(\kappa^2 + 2\right)\left(\kappa^8 + 8\kappa^6 + 21\kappa^4 + 20\kappa^2 + 12\right)}$$

*denote the approximate slope $c^{(\mathrm{LKS})}$ specialized to when $q = \mathcal{N}(\mu_q, 1)$. Then, for any $\mu_q \neq 0$, the function $\kappa^2 \mapsto c_1^{(\mathrm{LKS})}(\mu_q, \kappa^2)$ is strictly increasing on $(0, \infty)$. Further,*

$$\lim_{\kappa^2 \to \infty} c_1^{(\mathrm{LKS})}(\mu_q, \kappa^2) = \mu_q^4/2. \tag{5}$$

*Proof.* **Proof of Claim 1, 2**. Recall $\widehat{S_l^2} := \frac{2}{n}\sum_{i=1}^{n/2} h_p(x_{2i-1}, x_{2i})$. With $p = \mathcal{N}(0, 1)$, and $k(x, y) = \exp\left(-\frac{(x-y)^2}{2\kappa^2}\right)$, $h_p(x, y)$ can be written as

$$h_p(x, y) := \frac{e^{-\frac{(x-y)^2}{2\kappa^2}}\left(\kappa^2 - \left(\kappa^2 + 1\right)x^2 + \left(\kappa^4 + 2\kappa^2 + 2\right)xy - \left(\kappa^2 + 1\right)y^2\right)}{\kappa^4}.$$

By Theorem 6, $c^{(\mathrm{LKS})} = \frac{1}{2}\frac{\left[\mathbb{E}_q h_p(\mathbf{x}, \mathbf{x}')\right]^2}{\mathbb{E}_p\left[h_p^2(\mathbf{x}, \mathbf{x}')\right]}$ which mainly involves expectations with respect to a normal distribution. In computing the expectation $\mathbb{E}_{x' \sim q} h_p(x, x')$, the idea is to form the density for a new normal distribution by combining $\frac{1}{\sqrt{2\pi\sigma_q^2}}e^{-(x-\mu_q)^2/2\sigma_q^2}$ (the density of $q$) and the term $e^{-\frac{(x-y)^2}{2\kappa^2}}$ in the expression of $h_p(x, y)$. Computation of $\mathbb{E}_{x' \sim q} h_p(x, x')$ will then boil down to computing an expectation wrt. a new normal distribution.

It turns out that

$$\mathbb{E}_{x \sim q}\mathbb{E}_{x' \sim q}[h_p(x, x')] = \frac{\mu_q^2\left(\kappa^2 + 2\sigma_q^2\right) + \left(\sigma_q^2 - 1\right)^2}{\left(\kappa^2 + 2\sigma_q^2\right)\sqrt{\frac{2\sigma_q^2}{\kappa^2} + 1}} = S_p^2(q),$$

$$\mathbb{E}_p\left[h_p^2(\mathbf{x}, \mathbf{x}')\right] = \frac{\left(\kappa^2 + 4\right)\left(\kappa^4 + 4\kappa^2 + 5\right)\kappa^2 + 12}{\kappa^3\left(\kappa^2 + 4\right)^{5/2}}.$$

Computing $\frac{1}{2}\frac{S_p^4(q)}{\mathbb{E}_p\left[h_p^2(\mathbf{x}, \mathbf{x}')\right]}$ gives the slope.

**Proof of Claim 3**. The expression for $c_1^{(\mathrm{LKS})}$ is obtained straightforwardly by plugging $\sigma_q^2 = 1$ into the expression of $c^{(\mathrm{LKS})}$. Assume $\mu_q \neq 0$. It can be seen that $c_1^{(\mathrm{LKS})}(\mu_q, \kappa^2)$ is differentiable with respect to $\kappa^2$ on the interval $(0, \infty)$. The partial derivative is given by

$$\frac{\partial}{\partial\kappa^2}c_1^{(\mathrm{LKS})} = \frac{\left(\kappa^2\right)^{3/2}\left(\kappa^2 + 4\right)^{3/2}\left(7\kappa^8 + 56\kappa^6 + 166\kappa^4 + 216\kappa^2 + 120\right)\mu_q^4}{\left(\kappa^2 + 2\right)^2\left(\kappa^8 + 8\kappa^6 + 21\kappa^4 + 20\kappa^2 + 12\right)^2}.$$

Since for any $\mu_q \neq 0$, $\frac{\partial}{\partial\kappa^2}c_1^{(\mathrm{LKS})} > 0$ for $\kappa^2 \in (0, \infty)$, we conclude that $\kappa^2 \mapsto c_1^{(\mathrm{LKS})}(\mu_q, \kappa^2)$ is a strictly increasing function on $(0, \infty)$. By taking the limit, we have $\lim_{\kappa^2 \to \infty} c_1^{(\mathrm{LKS})}(\mu_q, \kappa^2) = \mu_q^4/2$. $\qquad\square$

We are ready to prove Theorem 7. Recall that $\sigma_k^2$ is the kernel bandwidth of $n\widehat{\mathrm{FSSD}^2}$, and $\kappa^2$ is the kernel bandwidth of $\sqrt{n}\widehat{S_l^2}$ (see Section 2). Recall Theorem 7:

**Theorem 7** (Efficiency in the Gaussian mean shift problem). *Let $E_1(\mu_q, v, \sigma_k^2, \kappa^2)$ be the approximate Bahadur efficiency of $n\widehat{\mathrm{FSSD}^2}$ relative to $\sqrt{n}\widehat{S_l^2}$ for the case where $p = \mathcal{N}(0,1), q = \mathcal{N}(\mu_q, 1)$, and $J = 1$ (i.e., one test location $v$ for $n\widehat{\mathrm{FSSD}^2}$). Fix $\sigma_k^2 = 1$ for $n\widehat{\mathrm{FSSD}^2}$. Then, for any $\mu_q \neq 0$, for some $v \in \mathbb{R}$, and for any $\kappa^2 > 0$, we have $E_1(\mu_q, v, \sigma_k^2, \kappa^2) > 2$.*

*Proof.* By Proposition 12, the approximate slope of $n\widehat{\mathrm{FSSD}^2}$ when $\sigma_q^2 = 1$ is

$$c_1^{(\mathrm{FSSD})}(\mu_q, v, \sigma_k^2) = \frac{\sigma_k^2 \left(\sigma_k^2 + 2\right)^3 \mu_q^2 e^{\frac{v^2}{\sigma_k^2 + 2} - \frac{(v - \mu_q)^2}{\sigma_k^2 + 1}}}{\sqrt{\frac{2}{\sigma_k^2} + 1} \left(\sigma_k^2 + 1\right) \left(\sigma_k^6 + 4\sigma_k^4 + \left(v^2 + 5\right)\sigma_k^2 + 2\right)}.$$

Theorem 10 states that the approximate efficiency $E_1(\mu_q, v, \sigma_k^2, \kappa^2)$ is given by the ratio $\frac{c_1^{(\mathrm{FSSD})}(\mu_q, v, \sigma_k^2)}{c_1^{(\mathrm{LKS})}(\mu_q, \kappa^2)}$ (see Propositions 12 and 13) of the approximate slopes of the two tests. Pick $\sigma_k^2 = 1$, and for any $\mu_q \neq 0$, pick $v = 2\mu_q$. These choices give the slope

$$c_1^{(\mathrm{FSSD})}(\mu_q, 2\mu_q, 1) = \frac{9\sqrt{3} e^{\frac{5\mu_q^2}{6}} \mu_q^2}{2\left(4\mu_q^2 + 12\right)}.$$

We have

$$
\begin{aligned}
E_1(\mu_q, v, \sigma_k^2, \kappa^2) &= E_1(\mu_q, 2\mu_q, 1, \kappa^2) \\
&= c_1^{(\mathrm{FSSD})}(\mu_q, 2\mu_q, 1) / c_1^{(\mathrm{LKS})}(\mu_q, \kappa^2) \\
&\overset{(a)}{\geq} c_1^{(\mathrm{FSSD})}(\mu_q, 2\mu_q, 1) / \left(\frac{\mu_q^4}{2}\right) \\
&= \frac{9\sqrt{3} e^{\frac{5\mu_q^2}{6}}}{\mu_q^2 \left(4\mu_q^2 + 12\right)} := g(\mu_q),
\end{aligned}
$$

where at $(a)$ we use $c_1^{(\mathrm{LKS})}(\mu_q, \kappa^2) \leq \mu_q^4/2$ from (5). It can be seen that for $\mu_q \neq 0$, $g(\mu_q)$ is an even function i.e., $g(\mu_q) = g(-\mu_q)$. The second derivative

$$\frac{\partial^2}{\partial \mu_q^2} g(\mu_q) = \sqrt{3} e^{\frac{5\mu_q^2}{6}} \left(25\mu_q^8 + 45\mu_q^6 - 45\mu_q^4 + 81\mu_q^2 + 486\right) / \left(4\mu_q^4 \left(\mu_q^2 + 3\right)^3\right) > 0.$$

To see that $\frac{\partial^2}{\partial \mu_q^2} g(\mu_q) > 0$, consider two cases of $\mu_q^2 \geq 1$ and $0 < \mu_q^2 < 1$. When $\mu_q^2 \geq 1$,

$$g(\mu_q) \geq \sqrt{3} e^{\frac{5\mu_q^2}{6}} \left(25\mu_q^8 + 81\mu_q^2 + 486\right) / \left(4\mu_q^4 \left(\mu_q^2 + 3\right)^3\right) > 0,$$

because $45\mu_q^6 - 45\mu_q^4 \geq 0$. When $0 < \mu_q^2 < 1$,

$$g(\mu_q) \geq \sqrt{3} e^{\frac{5\mu_q^2}{6}} \left(25\mu_q^8 + 45\mu_q^6 + 486\right) / \left(4\mu_q^4 \left(\mu_q^2 + 3\right)^3\right) > 0,$$

because $-45\mu_q^4 + 81\mu_q^2 \geq 0$. This shows that $g(\mu_q)$ is convex on $(0, \infty)$. The function $g(\mu_q)$ on $\mathbb{R}\backslash\{0\}$ achieves global minima at $\mu_q = \mu_q^* := \pm\sqrt{\frac{3}{10}\left(\sqrt{41} - 1\right)} \approx \pm 1.273$. This implies that

$$
\begin{aligned}
E_1(\mu_q, v, \sigma_k^2, \kappa^2) &\geq g(\mu_q) \geq g(\mu_q^*) \\
&= \frac{25\sqrt{3} e^{\frac{1}{4}\left(\sqrt{41} - 1\right)}}{8\left(\sqrt{41} + 4\right)} \approx 2.00855 > 2.
\end{aligned}
$$

$\square$

# I Known Results

This section presents known results from other works.

**Theorem 14** ([9, Theorem 2.2]). *If the kernel $k$ is $C_0$-universal [6, Definition 4.1], $\mathbb{E}_{\mathbf{x}\sim q}\mathbb{E}_{\mathbf{x}'\sim q}h_p(\mathbf{x},\mathbf{x}') < \infty$, and $\mathbb{E}_{\mathbf{x}\sim q}\|\nabla_{\mathbf{x}}\log\frac{p(\mathbf{x})}{q(\mathbf{x})}\|^2 < \infty$, then $S_p(q) = \|\mathbb{E}_{\mathbf{x}\sim q}\xi_p(\mathbf{x},\cdot)\|_{\mathcal{F}^d} = 0$ if and only if $p = q$.*

**Lemma 15** ([8, Lemma 1]). *Let $U$ be an open subset of $\mathbb{R}^d$. If $k$ is a bounded, analytic kernel on $U \times U$, then all functions in the RKHS associated with $k$ are analytic.[4]*

**Lemma 16** (Weyl's Perturbation Theorem [4, p. 152]). *Let $\lambda_j(A)$ denote the $j^{th}$ eigenvalue of a square matrix $A$. If $A, B$ are two Hermitian matrices, then*

$$\max_j |\lambda_j(A) - \lambda_j(B)| \leq \|A - B\|,$$

*where $\|\cdot\|$ denotes the operator norm.*

**Lemma 17** ([31, Lemma 21.2]). *For any sequence of cumulative distribution functions, $F_n^{-1} \xrightarrow{d} F^{-1}$ if and only if $F_n \xrightarrow{d} F$.*

## Footnotes

[4]The result of [8] considers only the case where $U = \mathbb{R}^d$. However, the same proof goes through for any open subset $U \subseteq \mathbb{R}^d$.