[Reviews · NeurIPS 2017]

Reviewer 1



%%% UPDATE: Thank you for your response, which has been read %%% This is a nice paper, which builds on work from ICML 2016. The main contribution is a more efficient (liner time) version of a previously explored approach to goodness-of-fit testing based on Stein's method. There are some small gaps, or areas where it could be improved, but the overall standard seems high enough for NIPS. Some minor comments: - The effect of the choice of J is not explored in any theoretical detail. I am still left unsure whether J=1 would be sufficient if p and q differ in several parts of the domain. - The power of the test is only asymptotically \alpha, so that it might not be fair to compare different methods in terms of rejection rate alone at finite sample size. For instance, it could be that FSSD-opt just rejects almost everything (i.e. the power is incorrect). For the RBM example it was claimed that all tests had power \alpha \approx 0.05, but (a) this is almost impossible to see in the figure (for \sigma_per = 0 the rejection rates in Fig 2b are too small to see visually from the figure if they are close to 0.05 or not), and (b) this was not checked for the Gauss-Laplace problem. - How was \grad \log p(x) computed in the RBM example? - More details of the gradient descent over {v_i} could be provided. For instance, when p and q are well-separated, is it numerically hard to locate the optimal v_i? - The comparison of FSSD-rand to FSSD-opt is fine, but would not a FSSD-QMC (quasi Monte Carlo) alternative also be possible (and more competitive than FSSD-rand)? - Is there any theoretical guidance on the choice of the test/train split? - In theorem 9 there is a typo: "exists a functions"

Reviewer 2



The authors have utilized the kernel Stein discrepancy in order to construct a new one-sample hypothesis test [i.e. test the null of whether a sample of points were iid from a distribution P]. Previous work that had accomplished this was expensive for large datasets: for a dataset of n datapoints, it was O(n^2) to compute the statistic. Other previous work had constructed a test statistic that was O(n) in computation time, but as it only used a small amount of the total information available, the authors show it did not have a high power as a test statistic for local alternatives. The authors of this paper have constructed a test statistic that is also linear in n to compute, but it substantially more powerful for local alternatives. They demonstrate this empirically and also theoretically by studying the Bahadur efficiency of the statistic. This novelty of this paper is that it finds a clever way to construct a linear [in computation time] test statistic that angles at approximating the kernel Stein discrepancy ||E_Q{T_P g*}||_{H^d} [here g* is the optimizing g, H^d is the direct sum of d RKHSes, and T_p is the Langevin operator] with instead something like sup_x ||g*(x)||_2. These ideas could be used outside of the hypothesis testing regime, i.e., one could studying whether the topologies induced on the space of probability measures are the same for the approximated kernel Stein discrepancy and the original one. Detailed Comments: L191: [nit] You've used zeta for T_P k already. L205: The formulations of hat{FSSD} in (2) do not make it obvious that the computation required is O(d^2*J*n). Perhaps you can cite/use the formulation from L122 so this is more obvious? L261: What exactly does the "for some v" in Theorem 7 mean? E.g., can this be for all v? From v drawn from nu? L295: In their experiments, Chwialkowski et al. (https://arxiv.org/abs/1602.02964) showed that the KSD with a Gaussian kernel experienced a substantial loss in power in higher dimensions (as seen also in your Fig. 2a), but in work by Gorham and Mackey (https://arxiv.org/abs/1703.01717), they showed that this power loss could be avoided by using an inverse multiquadric kernel due to its less rapid decay. Since the Gaussian and inverse multiquadric KSD running times are the same and the power of the Gaussian KSD is known to degrade rapidly in higher dimensions, the inverse multiquadric KSD may be a more appropriate "quadratic-time" benchmark. How does the power of FSSD compare with the inverse multiquadric KSD in your experiment?

Reviewer 3



The kernel Stein discrepancy goodness-of-fit test is based on a kernelized Stein operator T_p such that under the distribution p (and only under the distribution p), (T_p f) has mean 0 for all test functions f in an RKHS unit ball. The maximum mean of (T_p f) over the class of test functions under a distribution q thus provides a measure of the discrepancy between q and p. This quantity is known as the kernel Stein discrepancy and can also be expressed as the RKHS norm of a witness function g. In this paper, instead of estimating the RKHS norm of g using a full-sample U-statistic (which requires quadratic time to compute) or an incomplete U-statistic (which takes linear time but suffers from low power), the authors compute the empirical norm of g at a finite set of locations, which are either generated from a multivariate normal distribution fitted to the data or are chosen to approximately maximize test power. This produces a linear-time test statistic whose power is comparable to that of the quadratic-time kernel Stein discrepancy in simulations. The authors also analyze the test's approximate Bahadur efficiency. This paper is well-written and well-organized, and the proposed test dramatically improves upon the computational burden of other tests, such as the maximum mean discrepancy and kernel Stein discrepancy tests, without sacrificing too much power in simulations. I certainly believe this paper is worthy of inclusion in NIPS. Although this is not often a salient concern in machine learning, in more traditional statistical applications of hypothesis testing, randomized tests engender quite a bit of nervousness; we don't want a rejection decision in a costly clinical trial or program evaluation to hinge on having gotten a lucky RNG seed. For permutation tests or bootstrapped test statistics, we know that the rejection decision is stable as long as enough permutations or bootstrap iterations are performed. For the authors' proposed test, on the other hand, randomization is involved in choosing the test locations v_1 through v_J (or in choosing the initial locations for gradient ascent), and the simulations use a very small value of J (namely, J=5). Furthermore, in order to estimate parameters of the null distribution of the test statistic, a train-test split is needed. Which raises the question: are the test statistic and rejection threshold stable (for fixed data, across different randomizations)? It would be helpful to have additional simulations regarding the stability of the rejection decision for a fixed set of data, and guidelines as to how many train-test splits are required / how large J needs to be to achieve stability. A minor comment: line 70, switch the order of "if and only if" and "for all f".